# Antibody response to *Aedes aegypti* D7L1+2 salivary proteins as marker of aggregate vector exposure and correlate of dengue virus susceptibility

Lauren E. Bahr[1], Marco Hamins-Puertolas[2], Darunee Buddhari[3], Ivona Petzlova[1,4,5], Fabiano Oliveira[6], Stephen J. Thomas[1,4], Saravanan Thangamani[1,4,5], Adam T. Waickman[1,4]*, Kathryn B. Anderson[1]*

1 Department of Microbiology and Immunology, State University of New York Upstate Medical University, Syracuse, New York, United States of America, 2 Department of Medicine, University of California, San Francisco, California, United States of America, 3 Department of Virology, Armed Forces Research Institute of Medical Sciences, Bangkok, Thailand, 4 Institute for Global Health and Translational Sciences, State University of New York Upstate Medical University, Syracuse, New York, United States of America, 5 SUNY Center for Vector-Borne Diseases, State University of New York Upstate Medical University, Syracuse, New York, United States of America, 6 Laboratory of Malaria and Vector Research, National Institute of Allergy and Infectious Diseases, National Institutes of Health, Bethesda, Maryland, United States of America

* waickmaa@upstate.edu (ATW); andekath@upstate.edu (KBA)

## Abstract

*Aedes aegypti* mosquitoes transmit several arboviruses of public health importance. Among these is dengue virus (DENV), a flavivirus whose global infection rates continue to rise each year. With limited options available for preventing or treating DENV infections, mosquito control remains the most widely implemented strategy to combat DENV transmission. Due to the global distribution of DENV, which infects an estimated 400 million people per year, vector suppression practices vary drastically by country and/or region and even small differences in microenvironment can significantly impact vector abundance. There remains a significant need to better understand vector exposure rates at an individual level to disentangle vector exposure and arboviral infection rates. To this end, we have optimized a serologic assay to assess the abundance of antibodies directed against the mosquito salivary proteins AeD7L1+2 as a surrogate metric of vector exposure. Utilizing this assay, we found that anti-AeD7L1+2 IgG levels were unable to identify low levels of *Aedes* exposure in individuals with limited prior *Aedes* exposure, indicating they are unreliable markers of an individual's recent exposure to low levels of these vectors. However, antibody levels against AeD7L1+2 were robust in plasma samples from individuals living in *Aedes* endemic regions. These antibody levels reflected seasonal changes in *Aedes* abundance and exposure, indicating their potential for use as an aggregate marker of vector exposure. Additionally, we found that there were negative associations with anti-AeD7L1+2 IgG levels and age in our cohort. Interestingly, we also

**Data availability statement:** All data required for the reproduction of the analysis presented in this study are present in the manuscript or in the corresponding supplemental data files.

**Funding:** Funding for this research was provided by the State of New York (ATW, SJT) and the National Institute of Allergy and Infectious Diseases (NIAID) R01AI175941 (K.B.A.). The funders had no role in study design, data collection and analysis, decision to publish, or preparation of the manuscript. ATW, IP, ST, SJT, and KBA received salary support from the State of New York. LB, ATW, SJT, and KBA received salary support from NIAID.

**Competing interests:** The authors have declared that no competing interests exist.

found that lower titers of anti-AeD7L1 + 2 IgG correlated with higher infection burden in households. This finding has implications for the potential interaction between AeD7L1 + 2 proteins or anti-AeD7L1 + 2 antibodies and DENV during infection events that will require further study.

## Author summary

*Aedes* mosquitoes can transmit several globally impactful viruses, including dengue virus (DENV). Currently there are no widely efficacious vaccines or treatments against DENV, making prevention of exposure to mosquitoes an important tool in preventing disease. While many methods for mosquito control and personal protection are in use, we still do not understand the true burden of exposure, especially at an individual level. To this end we investigated the use of host antibody response to a specific mosquito salivary protein, AeD7L1 + 2, that is injected by *Aedes* mosquitoes when they feed. We found that robust antibody response to this protein is seen in populations living in Thailand, who are highly exposed to *Aedes*, but not in those with limited exposure to small numbers of *Aedes*. In highly exposed groups we observed an increase in antibody response in the rainy season, when *Aedes* exposure is highest in Thailand. Interestingly, we also observed that individuals in households with high numbers of serologically inferred DENV infections had lower levels of anti-AeD7L1 + 2 antibodies than those in households with no inferred DENV infections. These observations. Measuring human antibody response to *Aedes* salivary proteins may offer better insight into population level exposure.

## Introduction

*Aedes* mosquitoes, most notably *Aedes aegypti*, serve as a vector for several globally important arboviruses including dengue virus (DENV) [1,2]. There are an estimated 400 million DENV infections annually, with roughly 96 million of those infections resulting in symptomatic disease [1]. Despite decades of research and concerted control efforts around the globe, the public health burden of DENV is growing rapidly [3,4]. While the majority of DENV infections self-resolve without the need for medical intervention, a fraction of DENV infections can progress to severe dengue can be fatal without appropriate supportive care [5]. While two DENV vaccines are currently available, both products offer incomplete protection against infection by all DENV serotypes, especially in baseline seronegative vaccinees [6–8]. As there is also no direct-acting antiviral or monoclonal antibody available to treat or prevent DENV infections, the most common infection prevention strategy remains vector control [5,9,10].

While vector exposure mitigation remains the cornerstone of most dengue control efforts, microspatial differences in mosquito abundance and variation in

individual-level vector exposure can make it difficult to assess the impact of intervention strategies as well as an individual's risk of exposure and/or infection. Traditional strategies for quantifying mosquito densities – such as oviposition traps, composite indexes, and adult collection- are highly imperfect proxies for numbers of bites received, precluding accurate assessments of risks of arboviral infection at the individual level in addition to being time, cost, and labor intensive [11]. Quantification of human immunity to *Aedes aegypti* salivary components offers promise as a more direct assessment of vector exposure, which could help direct vector control measures and provide insight into an individual's potential risk of infection.

Most studies seeking to quantify human immunity to mosquito salivary proteins (MSP) as a surrogate marker of vector exposure have relied on quantifying the abundance of mosquito salivary protein (MSP) specific antibodies in serum/plasma. The use of whole mosquito salivary gland in the form of salivary gland homogenate (SGH) or salivary gland extract (SGE) as antigen in serologic assays has been utilized by multiple groups, with results indicating changes in anti-MSP antibodies that reflect seasonal changes in *Aedes* density regionally [12–14]. However, a limitation of this approach is that the antigenicity of SGE/SGH is not genus specific, with immunity directed towards these antigens exhibiting significant cross-reactivity with both *Aedes* and *Anopheles* exposure [14,15]. The antibodies directed against the peptide Nterm-34kDa have been demonstrated as an antigenic biomarker that correlate with *Aedes* exposure [16–19], although antibody titers against this peptide are generally low and difficult to accurately quantify [16–18,20]. Recent work has identified AeD7L1+2 as highly antigenic *Aedes* salivary proteins that elicit a robust antibody response in serum from individuals living in Cambodia who are endemically exposed to *Aedes* [19]. These proteins belong to the D7 family, a well-studied MSP which function as scavengers of biogenic amines and have been shown to bind DENV *in vitro* [21,22]. However, the kinetics of AeD7L1/AeD7L2 specific immunity after vector exposure – as well as the relationship of antibody titers to traditional metrics of mosquito abundance – remain unclear.

Using the AeD7L1+2 recombinant proteins described above our group sought to explore the dynamics of IgG and IgM antibodies against AeD7L1+2 as both proximal and aggregate markers of *Aedes* exposure. We found that in individuals exposed to a group of 5 or 10 *Aedes* mosquitos the antibody response to AeD7L1+2 was not measurably different before and after exposure, even after repeated feedings. In contrast, in a cohort of individuals living in an *Aedes* endemic region, anti-AeD7L1+2 IgG levels were robust and correlated with season of sample collection, with samples collected following the *Aedes*-intense rainy season exhibiting higher titers than samples collected prior to the rainy season [23]. Additionally, we found that in households experiencing high inferred infection rates individuals had significantly lower levels of anti-AeD7L1+2 IgG. This suggests that anti-AeD7L1+2 IgG levels can be used as an aggregate marker of *Aedes* exposure in some populations, and that antibodies against mosquito salivary proteins may play a protective role in vector mediated DENV infection.

## Results

### Anti-AeD7L1+2 IgG is more a more sensitive measure than anti-AeD7L1+2 IgM in both limited and chronic *Aedes* exposure scenarios

To evaluate AeD7L1+2 specific humoral immunity as a marker of *Aedes aegypti* exposure we first sought to quantify the IgG and IgM responses to recombinant AeD7L1+2 in plasma samples from individuals living in regions with either chronic or limited *Aedes aegypti* exposure. The chronic exposure group was comprised of individuals from Kamphaeng Phet, Thailand, where *Aedes aegypti* exposure is endemic. Individuals living in Syracuse, New York, with no recent or significant travel to *Aedes aegypti* endemic regions comprise the limited exposure group for this analysis. To compare the antibody response between these two groups we utilized an electrochemiluminescence immunoassay based on the MesoScale Discovery platform to quantify the abundance of AeD7L1+2 specific IgG and IgM isotype antibodies present in serially diluted plasma. This assay platform was selected due to its highly sensitive detection capability, large dynamic range, and ability to be multiplexed with multiple antigens. Using this system, we observed a robust and titratable AeD7L1+2

specific IgG signal in individuals living in areas with chronic *Aedes aegypti* exposure relative to individuals with minimal vector exposure (**Fig 1A**). Based on the results of the titration, a plasma/serum dilution of 1:1,000 was selected for all future analysis due to the signal difference between the exposure groups observed, specifically when measuring IgG (**Fig 1A**). The pooled samples diluted 1:1,000 from each group were evaluated for levels of anti-AeD7L1 + 2 IgG and IgM (S1 Fig). Individual plasma samples from both groups were diluted 1:1,000 and levels of anti-AeD7L1 + 2 IgG and IgM were measured and ROC analysis confirmed that IgG, but not IgM, was able to act as a specific and sensitive marker of *Aedes aegypti* exposure (**Fig 1B**). Unpaired Wilcoxon test statistics found significantly higher levels of anti-AeD7L1 + 2 IgG ($p = 1.037e-10$), but not IgM ($p = 0.8429$) in the chronically *Aedes* exposed group compared to the group with limited exposure (**Fig 1C**). This first step in analysis indicated that antibodies, specifically IgG antibodies, against AeD7L1 + 2 were high in samples from individuals living in Kamphaeng Phet, Thailand, chronically exposed to *Aedes aegypti* and low in samples from individuals living in Upstate New York, with limited *Aedes aegypti* exposure.

**Anti-AeD7L1 + 2 antibody titers are poor markers of acute *Aedes* exposure**

With the knowledge that the antibody response to mosquito salivary proteins AeD7L1 + 2 could be quantified – and that *Aedes* exposure history correlates with anti- AeD7L1 + 2 IgG titers - we next sought to evaluate the antibody response in a controlled, proximal exposure setting. To this end, a human-mosquito challenge study was performed in Syracuse, New York, in which individuals living in this low *Aedes aegypti* exposure region were challenged with either 5 or 10 female *Aedes aegypti* mosquitoes, once on study day 0 and again on study day 42 (**Fig 2A**). During both exposures over 60% of the mosquitoes had successful feeds with one exception (S2 Fig). Samples were taken routinely throughout the study, allowing for the immune response to these exposures over time to be measured. AeD7L1 + 2 MSD assay was used to measure levels of bound IgG and IgM in the plasma of subjects taken both prior to mosquito exposure and on study days 35 and 77 (**Fig 2B**). These samples were selected to assess the anti-AeD7L1 + 2 antibody titer 5 weeks after both the primary and secondary feeding, at which time it was predicted IgG antibody titers would have stabilized. A one-way ANOVA with Geisser-Greenhouse correction and a Dunnett's multiple comparisons test were used to analyze the change in titers between days 35 and 77 of the study relative to the pre-exposure sample taken at day 0. No significant change was observed in either groups anti-D7L1 + 1 IgG or IgM levels over the course of the study. To confirm that the mosquitos used in this challenge study were capable of eliciting an anti-AeD7L1 + 2 antibody response in mammalian hosts after feeding– and that the lack of signal observed in this analysis wasn't due to antigen mismatch between the challenge mosquitos and the antigen probes used in the serology assay, we performed an additional mosquito feeding study utilizing naïve mice exposed in a similar fashion to the human volunteers. In this study, naïve C57Bl/6 mice were twice exposed to approximately 100 *Aedes aegypti* mosquitos over the course of 51 days, with serum collected pre-challenge, after the primary challenge, and following the second challenge. Strikingly, robust anti-AeD7L1 + 2 IgG antibody titers were observed in the *Aedes* exposed animals, but only after the second feeding/exposure (S3 Fig). These data suggest that the robust anti-AeD7L1 + 2 titers observed in individuals living in *Aedes*-endemic regions is the aggregate result of high-level exposure to *Aedes* mosquitoes, and that the proximal challenge with a relatively small number of mosquitoes was unable to induce a measurable immune response to these proteins in our human-challenge study.

**Anti-AeD7L1 + 2 IgG levels correlate with age and season, but not entomological surveillance data, in an endemically exposed population**

In light of the robust anti-AeD7L1 + 2 IgG titers observed in individuals living in *Aedes* endemic regions we next sought to better understand the dynamics of these antibodies in a population with chronic, high levels of mosquito exposure. To answer this question, we utilized samples taken from a longitudinal, multigenerational family-based cohort study located in Kamphaeng Phet, Thailand, where *Aedes* exposure is high and DENV is endemic [1,24]. Enrollment in in the cohort

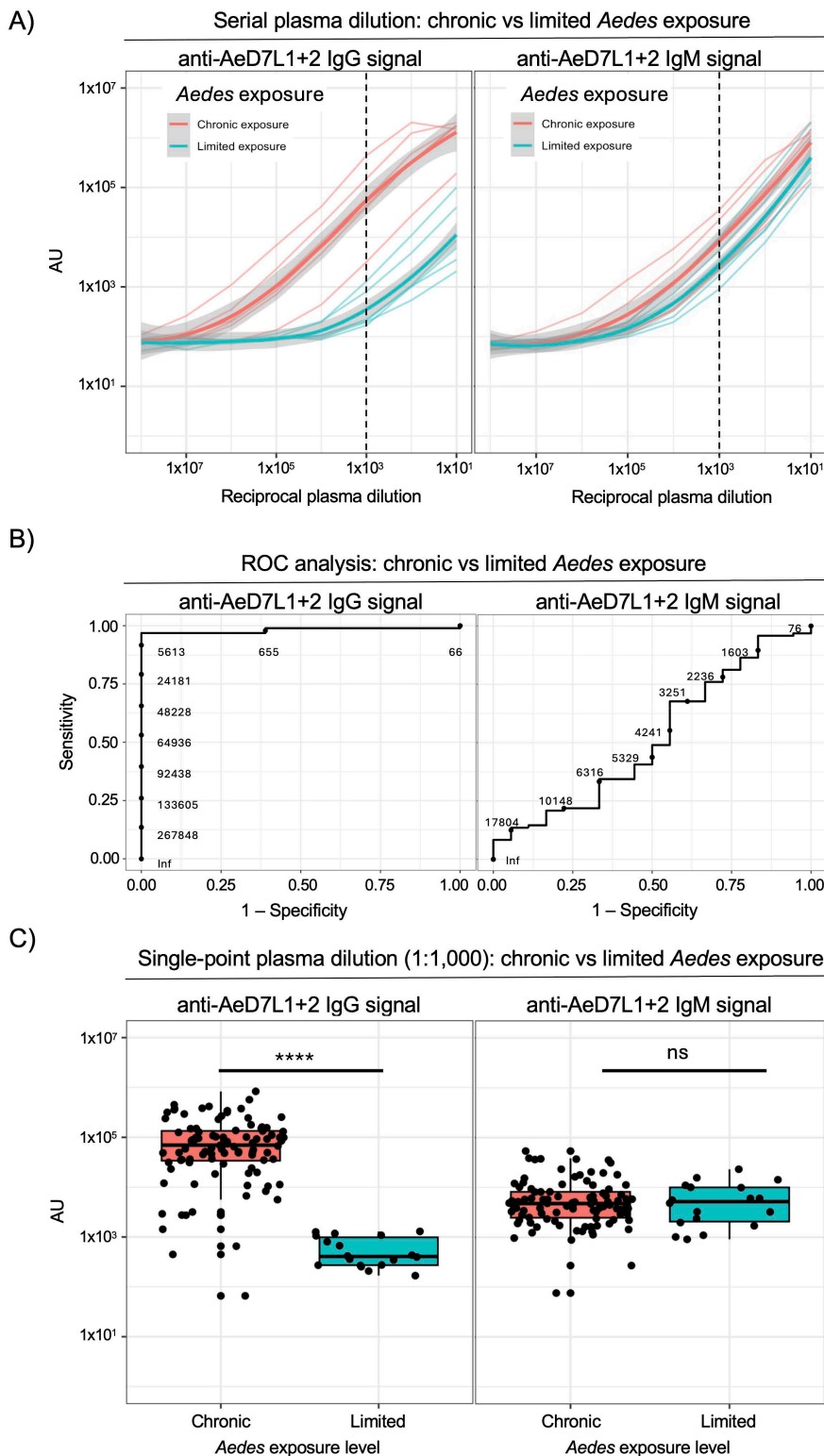

**Fig 1. Anti-AeD7L1+2 IgG is more a more sensitive measure than anti-AeD7L1+2 IgM in both limited and chronic *Aedes* exposure scenarios.**
A) Pooled plasma samples from a chronically *Aedes* exposed group (KFCS) and a limited *Aedes* exposure group were diluted 1:10 with seven ten-fold dilutions. AeD7L1+2 reactive IgG and IgM levels in the diluted plasma were measured by immunoassay. B) ROC analysis of the sensitivity of IgG and

IgM in delineating between *Aedes aegypti* chronically exposed (individuals living in Kamphaeng Pet, Thailand) and limited exposure (individuals living in Syracuse, New York) groups. C) Individual plasma samples from subjects in both chronic and low *Aedes* exposed groups were diluted 1:1000 and evaluated for anti-AeD7L1+2 IgG and IgM by MSD assay. Unpaired Wilcoxon test performed, IgG (p = 1.037e-10) IgM (p = 0.8429).

begins with a pregnant mother whose newborn is enrolled upon birth, giving a wide age range of subjects living in clustered households, and blood samples are collected at least annually from all enrolled members (S4 Fig). To investigate what, if any, relationship exists between anti-AeD7L1+2 IgG levels and volunteer age, a random selection of plasma samples from subjects in this cohort were evaluated by MSD assay. When analyzing the data using both a univariate model and a multivariate generalized additive model (GAM) to take season of collection, household-level random effects, and subject specific random effects into account, a significant negative relationship between age and anti-AeD7L1+2 IgG levels was observed (univariate p < 0.001, multivariate p < 0.001)(**Fig 3A**). To better understand if this decrease in anti-AeD7L1+2 IgG levels was a function of total antibody decrease with age we compared the anti-DENV HAI titers of the subjects with age at sample collection and found that these antibody titers increase with age (S5 Fig).

We next wanted to investigate the relationship between entomological burden and anti-AeD7L1+2 IgG levels. In addition to annual follow up blood sample collected from subject enrolled in the KFCS study; entomological sampling of households is periodically and randomly performed. Mosquitoes are collected at home visits using BG Sentinel traps and female *Aedes* are identified and counted. Plasma samples collected from members of a household taken within two months before or after entomologic collection were evaluated for anti-AeD7L1+2 IgG levels. No significant association between female *Aedes* count and anti-AeD7L1+2 IgG levels was observed when using univariate or multivariate models (univariate p = 0.661, multivariate p = 0.856) (**Fig 3B**).

Prior research has shown that human antibody response to mosquito salivary proteins or peptides trends with seasonal changes in vector exposure [20,18,19,25–27]. In Kamphaeng Phet, Thailand *Aedes aegypti* are most abundant during the rainy season, which lasts roughly from May to December. In this study seasons were defined as pre-rainy season (January-May), during rainy season (June-October), and post-rainy season (November-December). Anti-AeD7L1+2 IgG levels in these samples were measured and while univariate analysis found no relation between season and antibody level, multivariate analysis that accounted for subject age and household clustering found that there were significantly higher levels of AeD7L1+2 reactive antibodies in samples taken during the rainy season (p = 0.005), with reported geometric means trending higher in both rainy and post-rainy seasons (**Fig 3C**). These results reflect what has been observed previously in studies of antibody response to MSPs using SGH/SGE as well as specific peptides and proteins, indicating that the antibody response to recombinant AeD7L1+2 can be used as an aggregate measure of *Aedes* exposure.

## Lower levels of anti-AeD7L1+2 IgG are associated with more DENV seroconversions in enrolled households

Given the observation thus far that anti-AeD7L1+2 IgG levels in plasma appear to reflect aggregate *Aedes* exposure levels, we next wanted to investigate what, if any, relationship these antibody titers had to risk of DENV infection in individuals living in *Aedes* and DENV endemic regions of the world. To this end, we first sought to determine if there was any relationship between anti-AeD7L1+2 IgG levels and DENV-specific humoral immunity as assessed by DENV-specific hemagglutination inhibition (HAI) assay. Multivariate analysis that included subject age and accounted for household and subject specific random effects revealed no significant relationship between DENV-specific HAI values and anti-AeD7L1+2 IgG levels in samples evaluated by both HAI and MSD assays (p = 0.175) (**Fig 4A**). To better understand the relationship between DENV infection and anti-AeD7L1+2 IgG levels we wanted to compare the antibody response across households with varying numbers of inferred DENV infections over the course of a year. Households with zero to five seroconversions occurring among enrolled participants between annual samples were identified and anti-AeD7L1+2 IgG levels were measured. Interestingly, higher numbers of inferred infections had a significant relationship with lower

A)

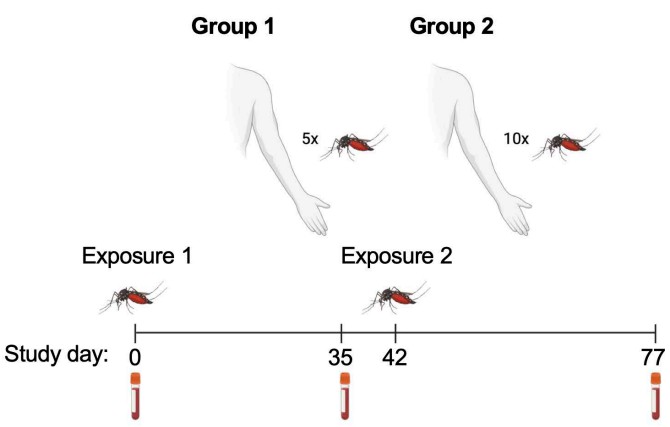

B)

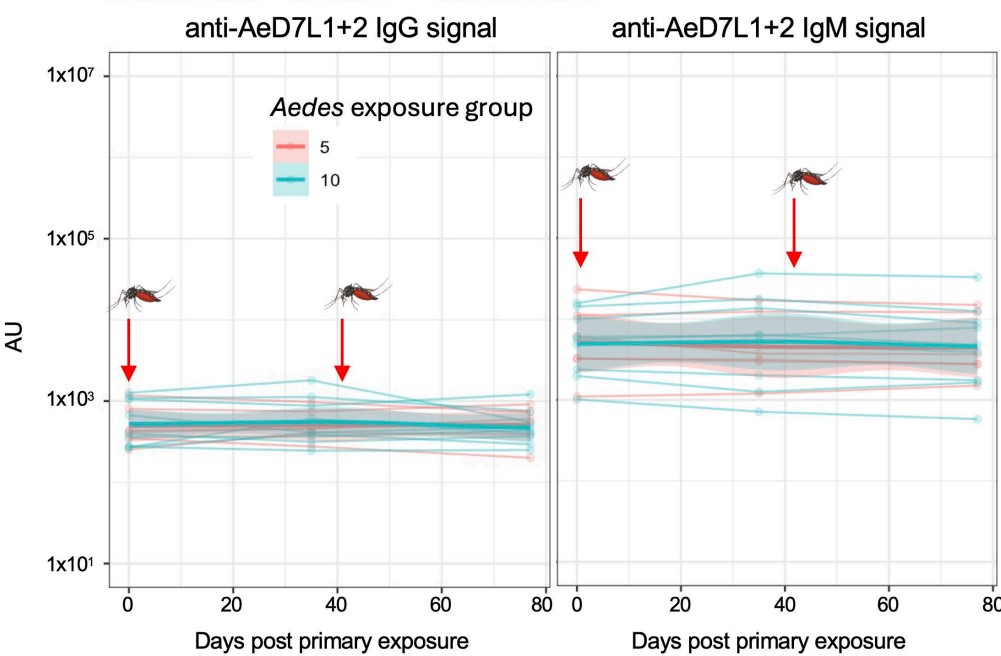

**Fig 2. Anti-AeD7L1+2 antibodies are not robust markers of acute *Aedes* exposure.** Schematic of human-mosquito challenge study (MOSQ). Figure Created in BioRender. Bahr, L. (2025) https://BioRender.com/2o64swx. *Aedes* naïve individuals were challenged with a group of either 5 or 10 starved, pathogen free female *Aedes aegypti* mosquitoes twice over the course of the 84 day study, once on day 0 and once on day 42. Blood samples were taken at baseline day 0 and 1, 2, 3, and 7 days after each mosquito exposure. Additional blood samples were taken throughout the study at time-points of immunological interest. B) Serum samples taken on day 0 prior to *Aedes* exposure, day 35, and on day 77 of the study were evaluated for IgG and IgM response to AeD7L1+2 for groups exposed to 5 or 10 *Aedes aegypti*. Arrows indicate *Aedes* exposure and gray shaded area indicates a 95% confidence interval. One-way ANOVA with Geisser-Greenhouse correction and a Dunnett's multiple comparisons test performed to compare titers at day 35 and day 77 to baseline sample at day 0. IgG 0-35 p=0.8504, 0-77 p=0.5412. IgM 0-35 p=0.5968, 0-77 p=0.9844.

A)

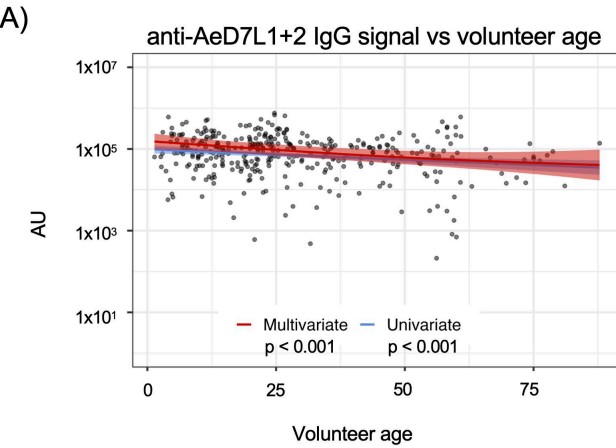

B)

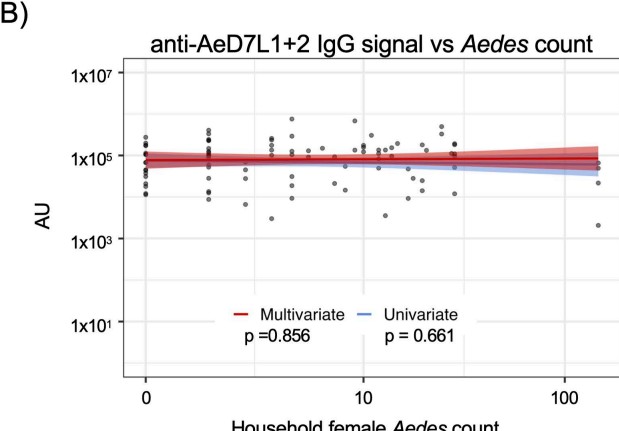

C)

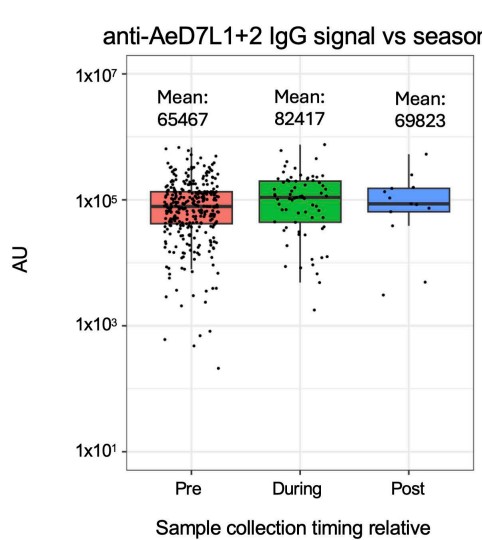

**Fig 3. Anti-AeD7L1+2 IgG levels correlate with age and season, but not entomological surveillance data, in an endemically exposed population.** A) Anti-AeD7L1+2 IgG levels stratified by age of subject at time of first available annual sample. Gray shaded area represents 95% confidence interval. There is a significant relationship with age in both univariate (p<0.001) and multivariate (p<0.001) models where older individuals have lower

---

levels of anti-AeD7L1 + 2 IgG. B) Households with both female *Aedes* collection data and a blood sample within a two-month period identified. Plasma samples collected within two months of entomological investigation identified and anti-AeD7L1 + 2 IgG levels measured by MSD. Gray shading represents 95% confidence interval. There is no significant relationship with log10 mosquito counts in either univariate (p = 0.661) or multivariate models (p = 0.856) C) Plasma samples stratified by the season in which they were collected: pre rainy season, during rainy season, or post rainy season. Values reported in the figure are the geometric mean for each collection season. No significant relationship between of anti-AeD7L1 + 2 IgG levels and collection season in the univariate model (p = 0.156), but during the wet season they are significantly higher (p = 0.005) in a multivariate model.

levels of anti-AeD7L1 + 2 IgG levels in a multivariate analysis that included subject age, collection season, and accounted for household and subject specific random effects (p = 0.022) (**Fig 4B**). We observed that households experiencing the highest number of inferred DENV infections had the lowest levels of anti-AeD7L1 + 2 IgG.

To better understand if household size (defined as the number of individuals residing within a home) has an impact on measured antibody levels we analyzed the antibody levels for households of varying sizes enrolled in the cohort (**Fig 4C**). When using a multivariate model that included subject age, collection season, and accounted for household and subject specific random effects we observed a negative relationship between household size and levels of anti-AeD7L1 + 2 IgG (p < 0.001). Both household size and the number of inferred infections remained significantly associated with decreasing levels of anti-AeD7L1 + 2 IgG when included in both multivariate and univariate models together (p < 0.001 and p = 0.016 respectively). Taken together these results suggest that larger households within this cohort may have lower levels of aggregate of exposure to *Aedes* than individuals living in smaller households, and that there is a relationship between risk of DENV infection and anti-AeD7L1 + 2 antibody titers. However, the mechanistic link – if any - between these observations is still unclear and warrants further investigation.

## Discussion

With no widespread efficacious vaccine against DENV infection or treatment for disease resulting from infection, vector control is an important tool to prevent the spread of DENV. While there are many vector control strategies in place they are often cost, labor, and time intensive practices. Additionally, the true burden of vector exposure in the context of DENV remains unknown as *Aedes* vector prevalence, microclimates, and vector/human interaction vary regionally. Previous work has identified the human immune response to mosquito salivary proteins as a potential method for measuring vector exposure [13,20,18,19]. Numerous methods have been used in the attempt to find a suitable proxy for exposure, from measuring antibodies against whole SGE to finding specific antigenic peptides that elicit a measurable antibody response [14,15]. Recently two D7L proteins that are specific to *Aedes* saliva have been reported as a new tool for evaluating exposure to *Aedes* [19]. We sought to further explore the capabilities of these proteins to measure exposure in both limited and endemically exposed populations.

We found that samples from individuals living in Kamphaeng Phet, Thailand, an *Aedes* endemic region, had high levels of anti-AeD7L1 + 2 IgG compared to individuals living in Syracuse, NY, a region with no *Aedes aegypti* exposure but some limited *Aedes albopictus* exposure [28–30]. ROC analysis revealed that IgG, but not IgM, was a sensitive and specific marker of AeD7L1 + 2 levels in plasma. In samples from individuals living in an *Aedes* non-endemic region who were challenged with 5 or 10 female *Aedes* we observed no significant increase in anti-AeD7L1 + 2 IgG or IgM levels over the course of the study, which included two mosquito challenges. This lack of measurable immune response indicates that anti-AeD7L1 + 2 antibodies are not able to act as a proximal measure of *Aedes aegypti* exposure in individuals living in a non-endemic location. This is especially clear when looking at anti-AeD7L1 + 2 antibody levels in individuals who are endemically exposed to *Aedes*, which were significantly higher than those of the challenge subjects. While anti-AeD7L1 + 2 may not be a reliable indicator of recent low level *Aedes aegypti* exposure in this population we did we did observe a statistically significant change in antibody levels in samples from endemically exposed populations that reflected seasonal changes in the *Aedes* population. This suggests that at a population level anti-AeD7L1 + 2 IgG levels

A)

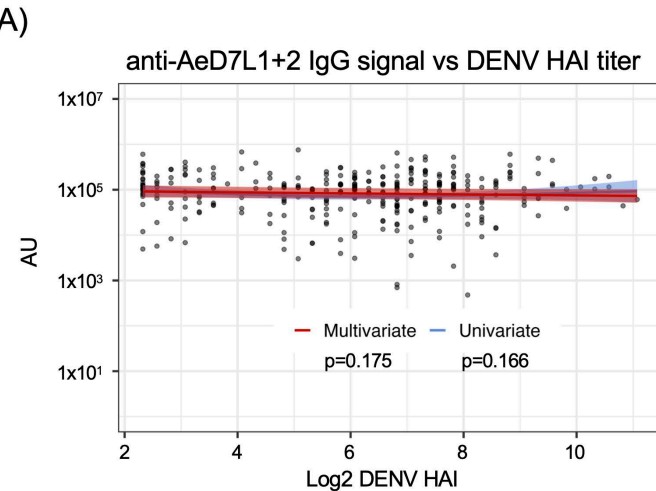

B)

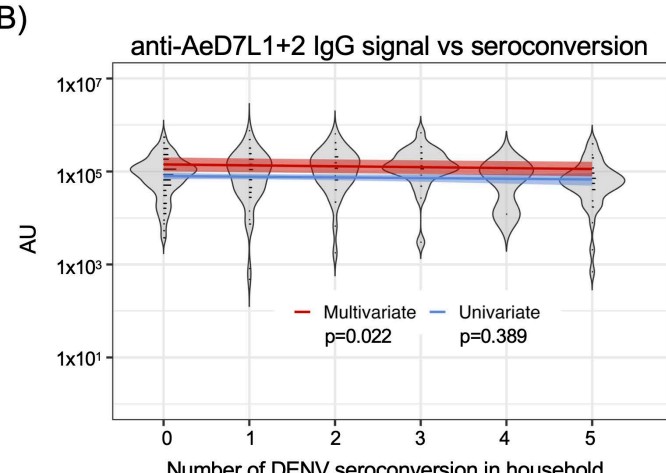

C)

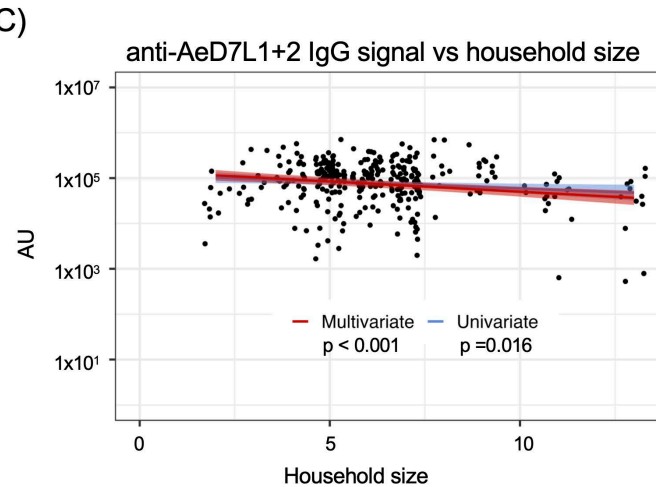

**Fig 4. Lower levels of anti-AeD7L1+2 IgG are associated with more DENV seroconversions in enrolled households.** A) Comparison of anti-DENV titers measured by HAI and the level of anti-AeD7L1+2 IgG measured by MSD assay in annual plasma samples. Gray shading represents 95%

confidence interval. There is no significant relationship between HAI titers and anti-AeD7L1+2 IgG levels in univariate (p = 0.166) or multivariate analyses (p = 0.175). B) Households with one to five seroconversions inferred by HAI titers between two annual samples were identified. Plasma samples from subjects in each household group taken after seroconversion were evaluated for anti-AeD7L1+2 IgG levels. There is no significant relationship between the values in a univariate (p = 0.389) model but there is in the multivariate (p = 0.022) models when including the household sample size. C) Anti-AeD7L1+2 IgG values analyzed by household size using both univariate (p = 0.016) and multivariate (p < 0.001) models.

can reflect aggregate changes in exposure but direct challenge with small groups of mosquitoes does not induce measurable antibody levels. Repeated exposure may be needed in order to induce and maintain these antibodies and further work to delineate how much exposure is necessary to induce a measurable antibody response is needed.

When exploring the dynamics of the anti-AeD7L1+2 IgG levels in individuals living with endemic *Aedes* exposure in Kamphaeng Phet, Thailand, we observed generally high titers that waned significantly with age. While the explanation for this trend is currently unclear, immunosenescnece, or the changes that occur to the immune system with age, could be one explanation for this observation. It has been shown previously that B cell progenitors decrease with age and that aging T cells express lower levels of costimulatory molecules needed to activate B cells, which could lead to lower levels of anti-AeD7L1+2 antibodies in older populations [31–33]. Additionally, this trend could be due to general waning immunity and/or immunological tolerance associated with age, although in this cohort we do not observe a decline in antibodies to DENV with volunteer age but instead an increase (S5 Fig).

When investigating the relationship between female *Aedes* collected and counted within a household and the anti-AeD7L1+2 IgG levels from subjects within those houses we found no significant association when using either a univariate or multivariate model. This could in part be due to the fact that *Aedes* mosquitoes are day feeders, so subjects who work or go to school outside of the household may be exposed to differing levels of *Aedes* [26], making collection within the household a helpful but imperfect reflection of vector exposure experienced by all household members [34]. The BG Sentinel traps used within this cohort are passive traps, designed to draw vectors in with olfactory attractants and convection currents that mimic those of a human host [35]. While the BG Sentinel has been shown to be an effective method for trapping *Aedes* it may not accurately reflect the bite rate experienced by those living in households with traps [36].

Previous work has shown that the immune response to MSPs reflect seasonal changes in *Aedes* exposure, making this a measure of aggregate exposure [25]. In Thailand rainy season occurs from January to May, and we sought to explore how season impacted anti-AeD7L1+2 IgG levels, as other have shown changes in *Aedes* adult and larval populations with seasons, although some seasonal definitions vary [29,30]. The cohort used in this study involves annual sampling with the bulk of the subjects enrolled pre-rainy season but due to asynchronous enrollment there are samples taken during and post-rainy season that allowed us to compare anti-AeD7L1+2 IgG levels in samples taken throughout the seasons. When using a multivariate model, we found that there were significantly higher anti-AeD7L1+2 IgG levels in samples taken during the rainy season. This follows observations previously published, demonstrating that higher levels of mosquito exposure during rainy season are reflected in higher levels of anti-MSP antibodies. This leads us to believe anti-AeD7L1+2 IgG is a suitable aggregate marker of vector exposure at the population level. These changes are difficult to observe at an individual level due to the variation between samples, making it difficult to ascertain individual proximal changes in exposure in this sample set as well. More work needs to be done in order to validate and optimize a marker of proximal, individual exposure. A potential marker for this use would need give a robust and reliable signal but also be sensitive enough to delineate between exposures.

Finally, we wanted to better understand what if any relationship anti-AeD7L1+2 antibodies had to DENV infections within mutigenerational households located in Kamphaeng Phet, Thailand. We first compared subject's annual anti-DENV HAI titers to their anti-AeD7L1+2 IgG levels and found there was no significant relationship between the two values. Given our observation that anti-AeD7L1+2 IgG was capable of identifying aggerate *Aedes* exposure, we hypothesized that subjects living in households with higher levels of DENV infection would have higher levels of anti-AeD7L1+2 IgG

when compared to subjects in households with little to no DENV infection. Interestingly we found the opposite to be true. Subjects from households with five DENV seroconversions over the course of one year had significantly lower anti-AeD7L1+2 IgG levels than those in households that experienced less seroconversions over a year, which remains true when accounting for household size. Furthermore, we observed that with increasing household size there were decreasing levels of anti-AeD7L1+2 IgG in this cohort. If we assume vector exposure levels are constant between household this could be a fractional effect, with the same level of vector exposure being spread between more individuals in larger households. This could be explored further by looking at the location of these households in relation to each other and by doing more target vector surveillance to better understand the level of exposure experienced by these households.

The inverse relationship between anti-AeD7L1+2 IgG levels and DENV seroconversions was an intriguing result of this study, leading us to consider what role these anti-MSP antibodies may have in the context of vector mediated DENV infection. Although this is an observation in a relatively small group, others have found evidence of anti-MSP antibodies playing a protective role in flavivirus infection. Previous work by Uraki, et al. has revealed that antibodies against AgBR1 abrogate the effects of vector delivered Zika virus (ZIKV), where mice treated with anti-AgBR1 antibodies before being infected via ZIKV infected *Aedes* had a longer mean survival time compared to the untreated group [37]. Further work in the context of DENV infection is needed in order to better understand the role of antibodies against MSPs in vector mediated infection. In the future we plan to evaluate the effect of antibodies against MSPs in the vector driven transmission of DENV, considering that in this study anti-D7L1+2 antibodies likely act as a proxy for other anti-MSP antibodies present. It is possible that other anti-MSP antibodies, or a combination of antibodies, may have a mechanistic effect in preventing DENV transmission.

While promising, some limitations of this study must be acknowledged. As mentioned previously, this study utilized samples from studies and cohorts with limited sample numbers and specific sampling schemas. This limited the timeframes with samples available to investigate, such as the lower number of samples taken during and post rainy season. This was accounted for using a multivariate model during analysis, but it would be interesting to measure the levels of anti-D7L1+2 antibodies in larger numbers of samples from those timeframes.

Overall, this study found that IgG in human serum or plasma directed towards AeD7L1+2 proteins acts as an aggregate marker of *Aedes* exposure in an endemically exposed population. These antibody levels were robust in individuals living in an *Aedes* endemic region yet significant changes in antibody response were observed that correlated with seasonal *Aedes* abundance. This tool could be useful in better predicting risk of exposure and DENV transmission at a population level moving forward.

## Methods

### Ethics statement

This study was approved by the Thailand Ministry of Public Health Ethical Research Committee; Siriraj Ethics Committee on Research Involving Human Subjects; Institutional Review Board for the Protection of Human Subjects, State University of New York Upstate Medical University; and Walter Reed Army Institute of Research Institutional Review Board (protocol number 2119). This study and all associated analysis were approved by the SUNY Upstate Institutional Review Board, with written informed consent obtained from all volunteers.

### Mosquito-human challenge study

A pathogen-free colony of *Aedes aegypti* maintained at the Vector Biocontainment Laboratory, SUNY Upstate Medical University was used in this study. The progenitors of the pathogen-free *Ae. aegypti* colony were received from the Wadsworth Center, New York State Department of Health; these mosquitoes were originally collected in Orlando, Florida, USA. Mosquitoes were reared in an insectary maintained at 27°C with 80% relative humidity and a 16-hour light/8-hour

dark photoperiod, as described previously [38]. Prior to human exposure studies, mosquitoes from our colony were screened via PCR to confirm the absence of select infectious pathogens, including DENV, Chikungunya virus, Zika virus, Yellow fever virus, Mayaro virus, West Nile virus, and Usutu virus. Three days before the study, 5 or 10 female mosquitoes (3–5 days post-eclosion) were sorted into each feeding capsule. The feeding capsule was prepared by shaving off the bottom portion of 50 ml Falcon tubes. The open end of the Falcon tube was then sealed with mesh. On the days of human exposure, feeding capsules containing 5 or 10 mosquitoes were placed on either the left or right arm of the volunteer and allowed to feed for 10 minutes. The feeding capsule was secured to the human hand with athletic tape. After feeding, mosquitoes were anesthetized, and the number of fed mosquitoes was counted and documented. One volunteer in the 5 *Aedes aegypti* exposure group was only exposed once on day 0 of the study. Volunteers for this exposure study were screened for severe hypersensitivity to insect bites and for recent travel to *Aedes* endemic areas.

## Kamphaeng Phet family cohort study

Plasma samples from individuals enrolled in a longitudinal cohort study based in Kamphaeng Phet, Thailand were used in this study. As described previously [23], enrollment began with a pregnant mother in her third trimester. If the mother lived in a multigenerational household, with her newborn, at least one other child, and an adult at least 50 years of age, the participants were briefed and enrolled. Blood samples were taken upon enrollment and then annually as routine follow up. Age, household location, and date of samples were recorded through the study. Entomological surveys were performed periodically as part of household sampling as previously described. Hemagglutination inhibition assays (HAI) were performed on samples collected during routine follow ups to measure anti-DENV antibody titers, as previously described by Anderson et al. [23], allowing for the capture of DENV seroconversions that occur between annual samples. For this analysis, HAI titers are shown as the geometric mean of individual DENV-1, -2, -3 and -4 HAI titers.

## Antibody detection using Meso Scale Discovery (MSD) assay

Serum and plasma were evaluated for presence of anti-D7L1+2 IgG and IgM using a U-plex Development Pack, 7-assay. This assay allows the development of a custom assay capable of multiplexing 7 target antigens in each well of a 96-well plate. Recombinant D7L1 (GenBank: AAL16049.1) and AeD7L2 (GenBank: AAA29347.1) antigens were biotinylated separately using an EZ-Link Sulfo-NHS-LC-Biotinylation Kit (ThermoScientific, cat# 21435) then mixed at a 1:1 ratio. Following MSD protocol plates coated with biotinylated AeD7L1+2 conjugated to MSD linker proteins were incubated for 1 hour at room temp shaking at 700 rpm. Plates were washed and then incubated with MSD Blocking Buffer A (cat #R93BA-2) at room temperature for 30 minutes shaking at 700 rpm. After washing the plate, serum or plasma diluted in MSD diluent 100 (cat #R50AA-4) was used to coat the plate. Serum/plasma coated plates were incubated for 1 hour at room temperature shaking at 700 rpm. Plates were coated with anti-IgG (MSD, cat #D21ADF) or anti-IgM (MSD, cat # D21ADD) secondary antibody after washing and incubated for 1 hour at room temperature shaking at 700 rpm. After a final wash MSD Gold Read Buffer B (MSD, cat #R60AM-2) was added to the plate and results were read on an MSD instrument. Readout values reported as electrochemiluminescent output. Raw data files were exported to MSD analysis platform and analyzed further in R and Graphpad Prism.

## *Aedes aegypti* mouse exposure study

Female BALB/C mice, aged between 6 and 10 weeks, were anaesthetized with ketamine/Xylazine and exposed to approximately 100 starved, pathogen free, female *Aedes aegypti* for 15 minutes. A second feed following this protocol was done 41 days after the first feed. Serum samples were collected from the mice before exposure and 10 days after each exposure to monitor for immune response to mosquito salivary proteins. Serum was evaluated using the MSD immunoassay described above using an anti-mouse IgG secondary antibody (MSD, cat # R32AC-1).

## Statistical analysis

Statistical analysis and data visualization was performed using R Statistical Software (R version 4.4.0), RStudio [39] (version 2024.12.0 + 467), and GraphPad Prism (version 10.3.0). Paired t-tests were used when comparing the log10 MSD values associated with two groups. Univariate and multivariate regressions were conducted on KFCS samples using the mgcv package [40], which allowed for the implementation of generalized additive models (GAMs) with a penalized thin plate regression spline for continuous variables of interest including age, and $\log_2$ geometric mean DENV HAI. A linear model was implemented for $\log_{10}$ transformed female *Aedes* count. Specifically, we implemented regression analyses to examine the relationship between MSD values and age, female *Aedes* counts, season of collection, DENV HAI titers, the number of seroconversions that occurred within an individual's household in the preceding interval and the total number of samples in said interval. All multivariate models accounted for age, season of collection in addition to individual and household random effects except for the multivariate analysis of $\log_{10}$ transformed female *Aedes* count which only included household random effects as each subject only had one relevant datapoint.

## Supporting information

**S1 Fig. Anti-AeD7L1 + 2 IgG and IgM levels in pooled plasma samples from chronic and limited *Aedes* exposed groups.** A) Anti-AeD7L1 + 2 IgG and IgM values for the pooled plasma samples from Fig 1A. B) ROC analyses of the sensitivity of these antibody levels in delineating exposure using pooled samples.
(PDF)

**S2 Fig. Feeding efficacy of Aedes aegypti in mosquito human challenge study.** Mosquito engorgement data for first and second Aedes aegypti exposures. Percentage of mosquitoes out of 5 or 10 that fed during each exposure.
(PDF)

**S3 Fig. Anti-AeD7L1 + 2 antibody levels increase in mice after two *Aedes aegypti* feeds.** 7 female BALB/C mice were fed on by approximately 100 *Aedes aegypti* mosquitoes twice, once on day 0 and once on day 41. Serum was collected before exposure and 10 days after each exposure. Values analyzed using Friedman's test and a Dunn's multiple comparisons test. Antibody levels increased significantly after the second *Aedes* exposure (p = 0.0066).
(PDF)

**S4 Fig. Kamphaeng Phet family cohort study.** Schematic of KFCS enrollment and sample collection. Figure Created in BioRender. Bahr, L. (2025) https://BioRender.com/2o64swx. Multigenerational households are enrolled starting with a mother from the household in their third trimester of pregnancy. Cord blood samples are taken from the newborn at birth and enrollment samples are taken from the household members. Longitudinal samples are taken annually.
(PDF)

**S5 Fig. HAI titers increase with age in KFCS samples.** Annual anti-DENV HAI titers compared to subject age at time of sample. Regression analysis represented by the solid black line, gray shaded area represents 95% confidence interval. Regression analysis suggests that there is a 0.05 (0.04-0.06) log2 unit increase in GMHAI titers for every year of age increase (p < .001).
(PDF)

**S1 Data. Supporting data for study.**
(XLSX)

## Acknowledgments

We gratefully acknowledge the members of the Institute for Global Health Institute (GHI) of SUNY Upstate Medical University for their support of these studies. Material has been reviewed by the Walter Reed Army Institute of Research. There is no objection to its presentation and/or publication. The opinions or assertions contained herein are the private views of the author, and are not to be construed as official, or as reflecting true views of the Department of the Army or the Department of Defense. The investigators have adhered to the policies for protection of human subjects as prescribed in AR 70–25.

## Author contributions

**Conceptualization:** Lauren E Bahr, Stephen J Thomas, Adam T. Waickman, Kathryn B. Anderson.

**Data curation:** Lauren E Bahr, Marco Hamins-Puertolas.

**Formal analysis:** Lauren E Bahr, Marco Hamins-Puertolas.

**Funding acquisition:** Darunee Buddhari, Adam T. Waickman, Kathryn B. Anderson.

**Investigation:** Lauren E Bahr.

**Methodology:** Lauren E Bahr, Marco Hamins-Puertolas, Darunee Buddhari, Ivona Petzlova, Stephen J Thomas, Saravanan Thangamani, Adam T. Waickman.

**Resources:** Fabiano Oliveira, Stephen J Thomas.

**Supervision:** Adam T. Waickman.

**Visualization:** Lauren E Bahr, Marco Hamins-Puertolas.

**Writing – original draft:** Lauren E Bahr, Marco Hamins-Puertolas.

**Writing – review & editing:** Lauren E Bahr, Marco Hamins-Puertolas, Stephen J Thomas, Adam T. Waickman.

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
