## [Decision Letter · Decision Letter 0]

14 Jul 2025

Antibody response to *Aedes aegypti D7L1+2 salivary proteins as marker of aggregate vector exposure and correlate of dengue virus susceptibility*

Dear Dr. Waickman,

Thank you for submitting your manuscript to PLOS Neglected Tropical Diseases. After careful consideration, we feel that it has merit but does not fully meet PLOS Neglected Tropical Diseases's publication criteria as it currently stands. Therefore, we invite you to submit a revised version of the manuscript that addresses the points raised during the review process.

Please submit your revised manuscript within 60 days Sep 12 2025 11:59PM. If you will need more time than this to complete your revisions, please reply to this message or contact the journal office at plosntds@plos.org. Please include the following items when submitting your revised manuscript:

We look forward to receiving your revised manuscript.

Kind regards,

Eric Calvo

Academic Editor

Audrey Lenhart

Section Editor

Shaden Kamhawi

co-Editor-in-Chief

Paul Brindley

co-Editor-in-Chief

*
**Journal Requirements:**
*

At this stage, the following Authors/Authors require contributions: Lauren Bahr, and Stephen J Thomas. Please ensure that the full contributions of each author are acknowledged in the "Add/Edit/Remove Authors" section of our submission form.

- TM on page: 13 line 406.

Note : Please ensure that the the figures captions are included in the manuscript file.

5) We have noticed that you have uploaded Supporting Information files, but you have not included a list of legends. Please add a full list of legends for your Supporting Information files after the references list.

Potential Copyright Issues:

i) Figures 2, and Supplemental figure 2. Please confirm whether you drew the images / clip-art within the figure panels by hand. If you did not draw the images, please provide (a) a link to the source of the images or icons and their license / terms of use; or (b) written permission from the copyright holder to publish the images or icons under our CC BY 4.0 license. Alternatively, you may replace the images with open source alternatives. See these open source resources you may use to replace images / clip-art:

7) We note that your Data Availability Statement is currently as follows: "All data required for the reproduction of the analysis presented in this study are present in the manuscript or in the corresponding supplemental data files". Please confirm at this time whether or not your submission contains all raw data required to replicate the results of your study. Authors must share the “minimal data set” for their submission. PLOS defines the minimal data set to consist of the data required to replicate all study findings reported in the article, as well as related metadata and methods (https://journals.plos.org/plosone/s/data-availability#loc-minimal-data-set-definition).

8) Please amend your detailed Financial Disclosure statement. This is published with the article. It must therefore be completed in full sentences and contain the exact wording you wish to be published.

2) If any authors received a salary from any of your funders, please state which authors and which funders.

9) Please provide a completed 'Competing Interests' statement, including any COIs declared by your co-authors. If you have no competing interests to declare, please state "The authors have declared that no competing interests exist". Otherwise please declare all competing interests beginning with the statement "I have read the journal's policy and the authors of this manuscript have the following competing interests:"

**
*Comments to the Authors:*
**

*
**Please note that one of the reviews is uploaded as an attachment.**
*

*
**Reviewers' Comments:**
*

Reviewer's Responses to Questions

*
**Key Review Criteria Required for Acceptance?**
*

**Methods**

-Are the objectives of the study clearly articulated with a clear testable hypothesis stated?

-Is the study design appropriate to address the stated objectives?

-Is the population clearly described and appropriate for the hypothesis being tested?

-Is the sample size sufficient to ensure adequate power to address the hypothesis being tested?

-Were correct statistical analysis used to support conclusions?

-Are there concerns about ethical or regulatory requirements being met?

*Reviewer #1: (No Response)*

*Reviewer #2: The assay details on the serological assay are missing (what sequences) - important since other groups reported the suitability of the D7 protein to determine recent exposures. Since the assay applied here does not have the resolution to determine recent exposures, the conclusions on susceptibility to DENV is not secured.*

*Reviewer #3: Are the objectives of the study clearly articulated with a clear testable hypothesis stated?*

Yes

Is the study design appropriate to address the stated objectives?

Yes.

Is the sample size sufficient to ensure adequate power to address the hypothesis being tested?

Partial

For the endemic cohort sample size appears reasonable but for the controlled mosquito challenge the sample size is small and may not have enough power to detect subtle antibody changes post exposure.

Were correct statistical analyses used to support conclusions?

Yes

Are there concerns about ethical or regulatory requirements being met?

No

*
**Results**
*

-Does the analysis presented match the analysis plan?

-Are the results clearly and completely presented?

-Are the figures (Tables, Images) of sufficient quality for clarity?

*Reviewer #1: (No Response)*

*Reviewer #2: The data do not support the conclusion and title of the manuscript (see detailed comments). The figure captions are missing making it harder to review the results. Key data such as changes in HAI titer as function of age are not presented. The data are log10 or log 2 transformed but the scales are always linear.*

Figure 2: The data demonstrate that the D7L1+2 “probes” used in this study (sequences were not provided) are not suited for detecting recent exposures. This is surprising since one of the co-authors contributed to a study demonstrating the suitability of D7L plate antigens in an ELISA-based assay (doi: 10.3389/fimmu.2024.1368066. PMID: 38751433). Is the discrepancy of the results due to the high sensitivity of the MSD platform (suboptimal signal-to-noise ratio)?

Figure 3: The correlation between age and antibody titers suffers from low sample size for the higher ages (n=7 based on Panel A). The authors discuss later that the lower antibody titers in the older participants may be due to immunosenescence. It would have been important to show a parallel figure with the HAI titers. Besides immunosenescence, there are other explanations such as induction of tolerance or, more likely, older individuals having lower activity levels outside the house and therefore lower chance for exposure.

Figure 4: The causality of seroconversion to DENV (no details provided whether this is solely based on HAI titers or whether other serology was established such as anti-NS1 tests) is not secured. It seems reasonable that DENV infected individuals have lower activity resulting in less exposure to mosquito bites. Since the antigenic probes using in this MSD-assay does not have sufficient resolution to determine recent exposure, the data cannot support the conclusions drawn by the authors.

*Reviewer #3: Does the analysis presented match the analysis plan?*

Yes

-Are the results clearly and completely presented?

Yes

-Are the figures (Tables, Images) of sufficient quality for clarity?

Yes

*
**Conclusions**
*

-Are the conclusions supported by the data presented?

-Are the limitations of analysis clearly described?

-Do the authors discuss how these data can be helpful to advance our understanding of the topic under study?

-Is public health relevance addressed?

*Reviewer #1: (No Response)*

*Reviewer #2: the conclusions are not supported by the data. causality is not demonstrated and the sample sizes for some of the age groups is low.*

*Reviewer #3: Are the conclusions supported by the data presented?*

Yes

Are the limitations of analysis clearly described?

Yes

Do the authors discuss how these data can be helpful to advance our understanding of the topic under study?

Yes

Is public health relevance addressed?

Yes

*
**Editorial and Data Presentation Modifications?**
*

*Reviewer #1: (No Response)*

*Reviewer #2: the supplementary data do not add additional information to the manuscript. Other details such as in the Materials and Methods and in the result section (e.g. HAI titers) would be important to add.*

*Reviewer #3: (No Response)*

*
**Summary and General Comments**
*

*Reviewer #1: The authors developed a serologic assay to detect IgG antibodies against Ae. aegypti D7 saliva proteins. This was done to better understand individual exposure and determine if this risk correlates with arboviral infection. The assay was unable to reveal when individuals were exposed to recent, low levels of mosquito bites, but was able to detect antibodies in individuals from endemic regions. Interestingly, the signal ebbed and flowed based on the mosquito biting season. There was also an inverse correlation between anti-D7 antibodies and infection burden in households.*

Major

-Is it possible that the sequence of D7 present in the laboratory-reared mosquitoes was not equivalent to the recombinant D7 and the D7 present in endemic Thailand? That would explain why the assay failed to detect anti-D7 antibodies in the New York population that was experimentally challenged with mosquito bites. Could these samples be analyzed for a broader target such as the less than ideal MSP? It’s hard to believe that there would be no seroconversion to any mosquito saliva target.

-Lines 268-279: More work/explanation is needed to confirm that the New York cohort didn’t seroconvert. Are the D7 sequences the same in the lab-reared mosquito as the recombinant protein? Mosquito saliva proteins evolve quickly and a single amino acid mismatch could have undermined the assay in this experiment. If the sequences are different, it would be helpful to run a separate assay either with updated D7 or a broader target like MSP to confirm that they didn’t seroconvert.

-Was hypersensitivity to experimental mosquito bites measured in the New York population? Those who were hypersensitive have seroconverted to mosquito saliva proteins in the past and should seroconvert again, although it’s possible that the Ae. aegypti D7 may activate B cells that recognize a similar but not identical D7 from North American mosquitoes – a type of original antigenic sin.

-Lines 244-245 and 322-332: Does the data confirm that there is a relationship between DENV infection and anti-D7 antibodies or is it possible that the assay is an indirect proxy for another anti-saliva response that influences DENV infection? How does DENV influence immunological memory? SARS-CoV-2, measles, etc. can negatively influence memory B cells. Is it possible that repeated DENV infection undermines the anti-mosquito saliva B cell population?

Minor

-The 2009 WHO classification system no longer includes DHF/DSS and instead focused on severe dengue.

-Is development of an IgG response linear or will overexposure of an antigen lead to a decrease in antibody production?

-On line 162 it suggests that the study continued through day 84 but the figure lists that it ends on day 77. Please clarify in the text or in the figure.

-Lines 201-204: What season is Ae. aegypti most abundant in this region? Please describe.

*Reviewer #2: Bahr et al report in their manuscript “Antibody responses to Aedes aegypti D7L1+2 salivary proteins as marker of aggregate vector exposure and correlate of dengue virus susceptibility” on the longitudinal assessment of serological responses to mosquito saliva antigens in samples from Thai Family Cohort study. The data demonstrate that the use of these particular plate antigens does not provide sufficient resolution to determine recent exposures, but allowed only assessment of cumulative exposures. The causality of serological responses to Aedes aegypti saliva proteins and dengue susceptibility is not demonstrated and therefore the data do not match the title of the manuscript.*

*Reviewer #3: Justify the choice of specific time points Days 35 and 77 in the human challenge, are these based on expected immune response?*

Briefly state how mosquitoes were screened to identify pathogen free status in the lab setting?

Exposure to only 5–10 mosquitoes might be insufficient, so justify in which basis the number was selected ?

The statement about anti D7L1+2 antibodies may be protective is too speculative for observational data.Clarify that anti D7L1+2 IgG serves better as a population level surveillance tool not an individual risk marker.

Include interpretation that lower antibody levels could reflect general immune immaturity or may be waning immunity.

Consider co-infections or household transmission chains as additional factors.

*PLOS authors have the option to publish the peer review history of their article (what does this mean? ). If published, this will include your full peer review and any attached files.*

**Do you want your identity to be public for this peer review?** For information about this choice, including consent withdrawal, please see our Privacy Policy .

*Reviewer #1: **Yes: ** Michael J Conway*

*Reviewer #2: **Yes: ** Elke Bergmann-Leitner*

*Reviewer #3: No*

**

**Figure resubmission:**

**Reproducibility:**



---

## [Decision Letter · Decision Letter 1]

21 Oct 2025

Dear Dr Waickman,

We are pleased to inform you that your manuscript 'Antibody response to *Aedes aegypti* D7L1+2 salivary proteins as marker of aggregate vector exposure and correlate of dengue virus susceptibility' has been provisionally accepted for publication in PLOS Neglected Tropical Diseases.

Best regards,

Eric Calvo

Academic Editor

Audrey Lenhart

Section Editor

Shaden Kamhawi

co-Editor-in-Chief

Paul Brindley

co-Editor-in-Chief

Reviewer's Responses to Questions

**Key Review Criteria Required for Acceptance?**

**Methods**

-Are the objectives of the study clearly articulated with a clear testable hypothesis stated?

-Is the study design appropriate to address the stated objectives?

-Is the population clearly described and appropriate for the hypothesis being tested?

-Is the sample size sufficient to ensure adequate power to address the hypothesis being tested?

-Were correct statistical analysis used to support conclusions?

-Are there concerns about ethical or regulatory requirements being met?

Reviewer #1: Yes

Reviewer #3: Revised manuscript is overall well conceived and scientifically sound. The study objectives are clearly stated and the multi-cohort design is appropriate for the research questions, leaving no major concerns.

**Results**

-Does the analysis presented match the analysis plan?

-Are the results clearly and completely presented?

-Are the figures (Tables, Images) of sufficient quality for clarity?

Reviewer #1: Yes

Reviewer #3: The results are clearly and systematically presented.

**Conclusions**

-Are the conclusions supported by the data presented?

-Are the limitations of analysis clearly described?

-Do the authors discuss how these data can be helpful to advance our understanding of the topic under study?

-Is public health relevance addressed?

Reviewer #1: Yes

Reviewer #3: well aligned with the data presented clearly supporting the claim that AeD7L1+2 IgG is a reliable population level marker of Aedes exposure while avoiding overstatement about its role in dengue susceptibility.

**Editorial and Data Presentation Modifications?**

Reviewer #1: (No Response)

Reviewer #3: (No Response)

**Summary and General Comments**

Reviewer #1: (No Response)

Reviewer #3: The authors have clearly responded to previous feedback.

PLOS authors have the option to publish the peer review history of their article (what does this mean? ). If published, this will include your full peer review and any attached files.

**Do you want your identity to be public for this peer review?** For information about this choice, including consent withdrawal, please see our Privacy Policy .

Reviewer #1: **Yes: ** Michael J Conway

Reviewer #3: No

---

## [Editor Report · Acceptance letter]

Dear Dr Waickman,

We are delighted to inform you that your manuscript, "Antibody response to *Aedes aegypti* D7L1+2 salivary proteins as marker of aggregate vector exposure and correlate of dengue virus susceptibility," has been formally accepted for publication in PLOS Neglected Tropical Diseases.

Best regards,

Shaden Kamhawi

co-Editor-in-Chief

Paul Brindley

co-Editor-in-Chief
